# MRI Deep Learning-Based Solution for Alzheimer’s Disease Prediction

**DOI:** 10.3390/jpm11090902

**Published:** 2021-09-09

**Authors:** Cristina L. Saratxaga, Iratxe Moya, Artzai Picón, Marina Acosta, Aitor Moreno-Fernandez-de-Leceta, Estibaliz Garrote, Arantza Bereciartua-Perez

**Affiliations:** 1TECNALIA, Basque Research and Technology Alliance (BRTA), Parque Tecnológico de Bizkaia, C/Geldo. Edificio 700, 48160 Derio, Spain; artzai.picon@tecnalia.com (A.P.); estibaliz.garrote@tecnalia.com (E.G.); aranzazu.bereciartua@tecnalia.com (A.B.-P.); 2Instituto Ibermática de Innovación, Unidad de Inteligencia Artificial Avenida de los Huetos, Edificio Azucarera, 01010 Vitoria, Spain; i.moya@ibermatica.com (I.M.); m.acosta@ibermatica.com (M.A.); ai.moreno@ibermatica.com (A.M.-F.-d.-L.); 3Department of Cell Biology and Histology, Faculty of Medicine and Dentistry, University of the Basque Country, 48940 Leioa, Spain

**Keywords:** deep learning, classification, Alzheimer’s, MRI, OASIS

## Abstract

Background: Alzheimer’s is a degenerative dementing disorder that starts with a mild memory impairment and progresses to a total loss of mental and physical faculties. The sooner the diagnosis is made, the better for the patient, as preventive actions and treatment can be started. Although tests such as the Mini-Mental State Tests Examination are usually used for early identification, diagnosis relies on magnetic resonance imaging (MRI) brain analysis. Methods: Public initiatives such as the OASIS (Open Access Series of Imaging Studies) collection provide neuroimaging datasets openly available for research purposes. In this work, a new method based on deep learning and image processing techniques for MRI-based Alzheimer’s diagnosis is proposed and compared with previous literature works. Results: Our method achieves a balance accuracy (BAC) up to 0.93 for image-based automated diagnosis of the disease, and a BAC of 0.88 for the establishment of the disease stage (healthy tissue, very mild and severe stage). Conclusions: Results obtained surpassed the state-of-the-art proposals using the OASIS collection. This demonstrates that deep learning-based strategies are an effective tool for building a robust solution for Alzheimer’s-assisted diagnosis based on MRI data.

## 1. Introduction

Back in 1907, Alois Alzheimer described how a 51-year-old woman died with severe dementia after four years of rapid memory degeneration. Alzheimer’s disease, named after him, is a dementia degenerative disease starting with mild memory impairment in the early stages and progressing to a complete loss of the mental and physical faculties [1].

The initial clinical manifestations of Alzheimer’s disease (AD) are difficult to define as there is a large variation between cognitive abnormalities, but they can be correlated to degeneration in some specific regions of the brain. At first, AD typically presents memory loss and poor judgement. Then, the disease makes the patient more and more dependent, and in its later stages, he/she requires constant supervision. Although there is currently no cure for AD, there is medication available that can temporarily reduce the symptoms, slowing down the progression of the disease and therefore, delaying the phase of complete dependence of the patient.

Over the years, different assessment tests have been proposed with the aim of diagnosing and monitoring the progress of AD.

The Mini-Mental State Examination (MMSE) exam [2], introduced in 1975, is the most extensively used questionnaire that examines six different modalities of patient cognitive ability. It is extensively used as it does not require special training or equipment and can be trustworthily used for the longitudinal monitoring of the AD progression. Unfortunately, its disadvantage is that it is affected by demographic, age and education factors and lacks sensitivity to measure progress in case of severe AD. With a maximum of 30 points, results can be evaluated as normal cognition, mild, moderate and severe stages of the disease. Different score ranges for each category have been defined and discussed in the literature throughout the years, and it has been expanded accordingly in its current form [3]. 

The Clinical Dementia Rating (CDR) [4], introduced in 1982, is an alternative structured interview for measuring dementia based on the scoring obtained in six different cognitive and behavioral areas that are combined to obtain a value between 0 and 3. The score values’ meanings are as follows: 0—cognitive normal, 0.5—very mild or questionable dementia, 1—mild dementia, 2—moderate dementia and 3—severe dementia. An initial limitation was the difficulty to distinguish mild and very mild cases from normal cognition, so the test was updated later on to introduce new aspects that reliably distinguish those cases [5]. 

Other tests are the Clock Drawing test (CDT), which is the second most used test, although it lacks sensitivity for identifying early or moderate AD, the Alzheimer’s Disease Assessment Scale (ADAS), the Global Dementia Scale (GDS) and the Neuropsychological Test Battery (NTB) [6].

However, definitive AD diagnosis relies on a magnetic resonance image (MRI) study [7]. MRI is a standard diagnosis technique free of radiation that is commonly used by healthcare providers for diagnosis, staging and follow-up of diseases. MRI scans are detailed three-dimensional anatomical images that can be used to examine almost any part of the body. In this case, the MRI technology makes it possible to recreate a 3D volume of the subject’s head in detail [8]. The role of MRI in AD diagnosis is that it allows studying brain structures and how they change over the time. In this sense, changes in hippocampus, frontal and parietal regions are evidential markers in the progress of the disease to dementia [9]. Other imaging modalities such as PET (positron emission tomography) are also complementarily used to study changes in brain functioning and activity of key proteins. In recent years, the diagnosis process of the disease is evolving and initiatives based on various identified biomarkers have been proposed, with recommendations of the International Working Group on how they should be used in the clinical practice [10]. 

In this work, we propose a new approach for the automatic diagnosis of Alzheimer’s disease by means of image processing and deep learning-based techniques using MRI data and CDR clinical annotation. Section 2 shows a review of available datasets and related work for comparison baseline definitions. Section 3 details the materials and methods used in this work and the proposed solution. Results are presented in Section 4 and discussion, conclusions and future research lines are presented in Section 5. Additionally, complementary information of materials and methods is included in Appendix B and Appendix C. 

## 2. Related Work

### 2.1. Data Collections

ADNI (Alzheimer’s Disease Neuroimaging Initiative) [11] is the most important and extensive collection of Alzheimer’s-related information, containing different types of images, such as MRI, PET, DTI (diffusion tensor imaging), genetic information, demographic information, cognitive tests and cerebrospinal fluid (CSF) biomarkers from over 1800 subjects. The collection has been gathered based on different studies: ADNI1, ADNI-GO, ADNI2 and ADNI3, starting in 2010 and ongoing up to date. Imaging information has been acquired over different follow-ups with the aim of developing biomarkers to track the progression and underlying changes happening in the presence of AD. MRI images from different sequences (structural, diffusion-weighted, perfusion and resting state) and PET can be found in raw format or post-processed. Data science challenges, such as TADPOLE (The Alzheimer’s Disease Prediction of Longitudinal Evolution) [12] in 2018, have made use of the ADNI collection, organizing the data in different standard sets containing longitudinal data (no longer available). Navigation in such a huge collection to obtain image data is a demanding task, and with the exception of the challenges, there are no standards sets that can be extensively used. For this reason, there are many research papers that use their own sets extracted from the original ADNI, making fair comparison of methodologies a very complex task, as reproductivity details to obtain a similar subset are not usually provided. 

OASIS (Open Access Series of Imaging Studies) [13] is another well-known initiative that has provided neuroimaging datasets openly available for research purposes. Three non-complementary, different datasets have been released over the years. These datasets are easily accessible, facilitating their usage and the validation of new proposed methodologies. OASIS-1 [14] (presented in 2007) is a cross-sectional collection of 416 subjects, where 100 are over 60 years of age and diagnosed with very mild or moderate Alzheimer’s (using CDR). The dataset includes a T1-weighted MRI scan for each subject, with the exception of 20 subjects without dementia, scanned twice on a subsequent visit for baseline comparison, and a list of annotations. The MRI data is offered processed, with skull removal operation and segmentation masks separating grey-matter (GM), white-matter (WM) and CSF performed, and whole-brain volume and estimated total intracranial volume (eTIV) calculated. OASIS-2 [15] (presented in 2010) is a longitudinal collection of 150 subjects scanned on different visits, including a total of 373 imaging sessions. T1-weighted MRI scans and annotations are provided for each subject, plus calculations of the whole-brain volume presented, however, data processed with the skull removal and segmentation operation are not available. Interestingly, 14 of the subjects were initially identified as non-demented and converted to demented in a later visit (CDR reference). Additionally, OASIS-3 [16] (presented in 2019) is a longitudinal collection of 1090 subjects, 650 non-demented and 493 at various stages of AD. The dataset includes both MRI (structural and functional sequences) and PET (metabolic and amyloid) images, with over 2000 and 1500 sessions respectively, including some post-processing as segmentation and PET analyses, and annotations. 

The ADCP (Alzheimer’s Disease Connectome Project) [17] has gathered data from 300 subjects, cognitively healthy and with dementia, over a timespan of 4 years. Collected data includes standardized demographics, MRI (structural, function, diffusion-weighted) and PET (only for some participants) imaging, behavioral information and clinical information (blood tests including genetics and CSF proteins’ analysis for 70% of the participants). The data follow standards and protocols defined in the Connectome Project [18]. The collection has not been published yet, although it is planned to be released in two steps, first the baseline collection and later the longitudinal collection. 

CADDementia (Computer-Aided Diagnosis of Dementia based on structural MRI data) [19] was an image processing challenge launched in 2014 containing 384 multi-center T1-weighted MRI images with AD, mild cognitive impairment (MCI) and healthy controls. A small training set (30 scans) with diagnosis labels was provided, although participants were encouraged to train on any suitable data (i.e., ADNI). The project provided a framework for the comparison of submitted computer-aided diagnosis methods. 

The MIRIAD (Minimal Interval Resonance Imaging in Alzheimer’s Disease) [20] dataset presented in 2013 included longitudinal volumetric T1 MRI scans of 46 subjects with mild–moderate AD and 23 controls, with a total of 798 scans acquired with the same equipment at established time intervals (0, 2, 6, 14, 26, 38 and 52 weeks, and 18 and 24 months). Additional information is available, such as gender, age and MMSE scores. 

The NACC (National Alzheimer’s Coordinating Center) database [21] was released in 2007, presenting uniform data collected from 29 centers over 30 years. It contains more than 900 data elements grouped in different datasets and metadata with 68 data elements (such as race, education, gender, diagnosis, stroke, depression, availability of DNA, availability of tissue, availability of MRI, etc.). As suggested by the metadata, imaging information is not available for all users. Besides, access to the database is controlled via four users’ types, from public to personnel restricted. 

### 2.2. Automatic Analysis

A systematic review of machine learning classification methods for assisted diagnosis is provided in [22], including literature works (conference and ArXiv are excluded) published during the years 2006–2016. Most works use traditional image processing methods for image pre-processing and feature extraction and traditional machine learning classification methods. Only 2 out of the 111 relevant studies considered were making use of a deep learning-based approach. Since 2016, as mentioned in [22], there has been a significant increase on the number of publications (with a big presence in conferences) using deep learning-based methods for the automatic classification of Alzheimer’s disease on MRI images. In this sense, [23] provides the first review focused on deep learning (DL) methods, however, not only for AD classification but also for other brain-based disorders. The changes in the methodologies can also be clearly observed in the most recent review study [24], that includes works in the 2005–2019 period. The analysis is mainly organized in three categories, including support vector machines (SVM), artificial neural networks (ANN) and DL, although others are discussed. The problem of the clinical interpretability of DL models, usually seen as black boxes, in comparison with the other strategies, is addressed in the conclusions section.

Other works, such as [25], provide an alternative extensive review of methods and strategies up to 2016, where different image modalities are included, not only MRI. They include an interesting critical review of different aspects in the discussion section. First, they present aspects affecting works’ comparison, such as the length of the follow-up period of patients in the datasets, characteristics of the population included, possible impairment in the data, evaluation metrics used and of other factors that can affect the performance of classification algorithms. Then, challenges still present on the AD classification works that are also common problems in the machine learning world, such as the ability to generalize, the size of datasets and samples, the reproductivity of the results and the heterogeneity of the AD disease, are presented. 

In the current work, the authors have decided to use the OASIS collection datasets as baseline validation of the proposed methodology. The OASIS datasets’ images and annotations are well-structured and organized, facilitating the reproductivity of the proposal and fair comparison of methods using exactly the same data. Other datasets provide more extensive data, but to the authors’ knowledge, there are no standard subsets that can be used for fair comparison, hence works tent to create their own subset and details to facilitate reproductivity are usually missing. For this reason, from now on, analysis of the state-of-the-art has been focused only on those works making use of OASIS for comparison purposes, although there are additional methods developed over the ADNI or other databases with an extensive related literature. Besides, to be more precise and facilitate reading, only the most relevant and reproducible methods are mentioned or described in detail. 

In general, there are two types of DL approaches: to combine Convolutional Neural Network (CNN) with traditional image processing and machine learning methods (for feature selection and/or classification) or to develop full end-to-end CNN DL solutions. With respect to combined approaches, the most interesting approximations are found in [26,27,28]. References [26,27] report accuracies close to 99%, whereas [28] reaches an 86.81% average accuracy for all classes. The reported accuracies with combined methods are impressive, however, in [26,27], few details are provided about the selection of cases in the dataset, meaning that the reported results can be biased and unrealistic. The proposal in [28] is interesting and reproducible, although no balanced accuracy (BAC) metric is provided, which is a more realistic metric to evaluate the accuracy of the approach in the presence of the high data imbalance of this dataset. Other metrics provided are accuracy, precision, recall and specificity, reporting 0% precision and recall for mild dementia and moderate AD classes due to the small number of cases in the dataset.

On the other side, with respect to full DL solutions, Reference [29], after testing different alternatives, proposed a 2D architecture which consists of an ensemble of three homogeneous and slightly different models containing convolutional, batch normalization, Rectified Linear Unit (ReLU) and pooling operations. For data augmentation, they proposed a cropping strategy, where three crops of size 112 × 112 pixels are extracted from each database sample, one from each image plane. They made use of the full OASIS-1 dataset, dedicating 70% for training, 10% for validation and 20% for testing. The model was trained independently and the ‘softmax’ classification layer with cross entropy was added to solve a four-class classification problem considering the CDR information on the dataset. A cost-sensitive training strategy is used for dealing with data imbalance, using a cost matrix to modify the output of the last layer of the networks for giving more importance to the underrepresented classes, assigning weights dependable to the number of samples of each class. Besides, models were optimized with the Stochastic Gradient Descent (SGD) [30] algorithm and early stopping regularization. Then, individual models’ answers are ensembled using a majority voting strategy, where the class with the majority of the votes is assigned as the output answer. The mean accuracy of the proposed architecture is 93.18%, with 94% precision (67% for mild dementia and 50% for moderate dementia) and 93% mean recall (33% for very-mild dementia and 50% for moderate dementia). 

More recently, Reference [31] provided a comprehensive review and comparison of methods proposed in the latest years. The authors mention the difficulties for fair comparison due to differences on participant selection, image pre-processing, model selection, validation procedure or the lack of implementation details (in more than half of the papers analyzed), which makes it difficult to reproduce the methods and could indicate biased performance of the reported metrics. They classified the identified approaches in four different groups considering the type of processing of the images: 2D slice-level, 3D patch-level, ROI-based and 3D subject level. Then, they have extended their open-source framework for the comparison of the selected CNN architectures, where they have detected that in general, neither of the proposals outperform an SVM with voxel-based features. As a result of their analysis, they provided different tables with the summaries of the studies analyzed and reported performances metrics (with data leakage identified) and the results of experiments implemented in their framework. In their experiments, they report the BAC metric and values obtained in five different executions (folds), comparing different classification architectures, training data configurations, image pre-processing strategies, intensity rescaling, training approaches, transfer learning approaches and classification tasks. For the OASIS collection (only patients over 62 years), they report average BAC within the range [0.61, 0.73] for the different experiments’ configurations (details in Table 6 in [31]). The best performance reported in their experiments uses the ADNI dataset and reaches 0.89 BAC. This reference offers a valuable and fair framework for the comparison of the results of new proposals. 

Considering the full DL solutions, possibly the most remarkable results are provided in [29], which makes use of 2D-based models (3 model resemble architecture), fair use of the complete OASIS-1 dataset, data augmentation and imbalance strategies and good learning practices, and they obtained 0.93 main accuracy in a four-class classification problem. However, the need of such architecture could be questionable as perhaps the results could be determined by the combination of ROIs from the 3 planes. 

## 3. Materials and Methods

### 3.1. Dataset

This work makes use of the OASIS collection for the evaluation and comparison of the proposed methodology with previous works, and of the OASIS-1 and OASIS-2 datasets.

OASIS-1 [14] is a cross-section open dataset containing information from 416 subjects aged between 18 and 96 years, acquired with a 1.5 T scanner. Out of the 416 entries, 20 are from subjects without dementia, imaged in a subsequent additional visit later to their initial visit, as a control group for ensuring reliability of the data and analysis provided. All the subjects have right-hand dominance.

The clinical status of the patient is established by the CDR scale, although the MMSE scale and other clinically relevant information along with demographic information is also provided for each entry. The annotation file contains the following information: ID, M/F, Hand, Age, Education (Educ), Socioeconomic status (SES), MMSE, CDR, estimated total intracranial volume (eTIV), normalized whole-brain volume (nWBV), atlas scaling factor (ASF) and Delay. ‘Educ’ indicates the years of education, SES uses the Hollinshead index of Social Position [32], MMSE score [2] and CDR scale [4] were established after medical examination, eTIV [33] and nWBV [34] were calculated as standard methods to analyze anatomical characteristics of the brain in the MRI images, ASF [33] was computed to transform the brain and skull from native space to the selected target atlas and ‘Delay’ for the 20 cases subgroup indicates the days after the first visit (mean: 20.55 days). The nWBV calculation requires a previous segmentation operation that separates WM, GM and CSF that has been performed based on estimations by the Markov random field model and further manual corrections.

Images of the OASIS-1 dataset are offered post-processed in 16-bit Analyze 7.5 format [35], with facial features out of the cranial values masked out, co-registered and converted to the Talairach and Tournoux standard atlas space [36] and inhomogeneity intensity due to the corrected magnetic field [37]. Additionally, different versions of the images are available: in RAW, processed without skull and segmented. In this work, we use processed images without skull, identified as “_t88_masked_gfc”. This makes the OASIS-1 dataset able to be widely used. Nevertheless, out of these ideal dataset conditions, it is worth validating the methodology and performance of the models over other raw datasets. 

OASIS-2 [15] is a longitudinal dataset of 150 subjects with ages between 60 and 96 years, scanned on various visits with a 1.5 T Vision scanner and at least one year difference, and a T1-weighted sequence, collecting a total of 373 imaging sessions. All the subjects have right-hand dominance. Interestingly, 14 of the subjects were characterized as non-demented on the initial visit and as demented afterwards, which can lead to studying the change on the whole-brain volume and structures to detect atrophy. Various subjects in the dataset were also part of the cross-sectional OASIS-1 (with different random identifiers), so the two datasets are not complementary.

The clinical status of the subjects is established by the CDR scale and contains the following annotations: Subject ID, MRI ID, Group, Visit, MR Delay, M/F, Hand, Age, EDUC, SES, MMSE, CDR, eTIV, nWBV and ASF. MRI ID identifies the number of the scan performed on the subject. ‘Group’ is a new annotation that classifies the subjects as ‘Nondemented’, ‘Converted’ or ‘Demented’, considering their status at the end of the study with respect to the beginning. Visit indicates the number of the visit of the entry and MR Delay indicates the number of days since the previous visit. 

Images of the OASIS-2 dataset are offered converted to 16-bit NiFTI1 format, corrected for inter-scan head movement and converted to the Talairach and Tournoux standard atlas space [36] with a rigid transform. Images are co-registered with a 12-parameter affine transformation and intensity inhomogeneity, and variations among contiguous regions are corrected. Unfortunately, only the raw images are provided, and skull removal or segmentations are not available. This makes it difficult to directly use this dataset, requiring some pre-processing, which is explained in Appendix C.

The number of samples included in each dataset detailed by CDR diagnosis is summarized in Table 1.

### 3.2. Data and MRI Volume Processing: Model Input

Images from both datasets have been labeled to face a classification problem. On the one hand, in two different classes, with subjects with CDR = 0 being part of one class and subjects with CDR = 0.5, CDR = 1 or CDR = 2 part of the other class. The reason for this classification is that only 2 examples of CDR = 2 and 28 of CDR = 1 are part of the OASIS-1 dataset, being such an under-representation that it would lead to inaccurate multi-class classification results. On the other hand, they were grouped in a three-class problem (as explained above, there are not enough examples of CDR = 2 for a four-class problem), where subjects with CDR = 0 are part of one class, subjects with CDR = 0.5 are part of another class and subjects with CDR = 1 and CDR = 2 are part of the remaining class.

All the available entries of the OASIS-1 dataset are used in this work. From the OASIS-2 dataset, some samples have been excluded due to skull removal problems, as explained in Appendix C and listed in Appendix A. No filtering per age (e.g., older than 60 years [31,38]) or other criteria has been performed in either of the datasets.

Training, validation and test sets are automatically randomly generated for the OASIS-1 and OASIS-2 datasets, separately. Each set contains a balanced number of samples per sex (M/F) to eliminate possible bias. The reason is that previous studies state that incidence is different in women and men [39,40], and that differences in the brain volume-associated measures are observed for male subjects [41] with an impact on the prevalence of the disease. Additionally, for the OASIS-2 (longitudinal) dataset, the different entries (MRI scans in different visits) from the same subject are always included in the same set. 

Only the horizontal (transverse) plane images have been used for training and testing the models. Some preliminary experiments were also performed using the frontal (coronal) plane images, but they were discarded as the results obtained were considered worse. 

Datasets of MRI volumes have original dimensions of 176 × 208 × 176, meaning that they contain 176 slices/images of 176 × 208 pixels size. Two different strategies have been considered in the experiments to include the MRI information. One strategy has been to use the whole 3D volume, which is especially interesting for models constructed with 3D convolutional layers. The other strategy has been to only use a limited number of slices of the volume, which has been demonstrated to be more interesting when experimenting with models built with 2D convolutional layers. Considering the center slice to be 88 (half of 176), an N number of slices from each side is extracted. Various experiments were executed with the same model with a different number of slices (10, 20, 30 and 50) to conclude that the optimum value was 10 slices. Increasing the number of slices demonstrated that no improvements of the results were obtained, and even that using ≥30 slices was unfavorable for the results. This suggests that the 10 center slices contain the most relevant information for diagnosis purposes. Then, original slices/images (176 × 208) have been resized to two different sizes depending on the model used, and implicit size restrictions if previous weights are used (e.g., ImageNet [42]). In this sense, two image sizes have been considered: 176 × 176 and 224 × 224 pixels.

Regardless of whether the model uses 2D or 3D convolutional layers, it can be trained with 3D data. In this sense, this work makes a comparison of 2D- and 3D-based architectures for the classification of Alzheimer’s CDR and provides a comparison of results, including additional improvement strategies. Additionally, an approach that converts the 3D data to 2D prior to training has also been implemented and included in the comparison. When data is converted to 2D, image slices are extracted from the 3D volume and treated independently during training, validation and testing. It is ensured that all image slices from the same sample are kept in the same data subset. Then, during the testing process, all images are evaluated independently, and a mean prediction is obtained (all images have the same weight), which determines the final prediction class.

Considering the data imbalance of the entries of the OASIS-1 and OASIS-2 datasets (especially in the first case), the network generator is configured to always include the same number of samples of each class in the current batch, independently if targeting a two-class problem or a three-class problem. To increase variability and avoid overfitting, the data imbalance strategy is complemented with a data augmentation strategy, where images are randomly vertically or horizontally flipped or rotated. When the data are treated as 3D volume, the same operations are randomly applied to all the slices in the volume. On the contrary, when the data are treated as 2D, different operations can be applied in the data augmentation process to each of the individual images of the volume.

#### 3.2.1. Data Normalization

Additionally, image values are normalized, and two strategies have been explored for image quality enhancement. In the basic form, 3D volume images are normalized to the [0–1] range. In an alternative configuration, and inspired by [43], a contrast stretching operation is performed. The 2nd and 98th percentile values of the volume are calculated and assigned as min and max values, and then the whole volume is normalized to the [0–1] range. 

#### 3.2.2. Metadata

Following a similar approach to what is proposed in [28], it has been explored whether sex and age metadata can benefit model performance. As analyzed in Appendix B, from all the metadata available in the OASIS-1 collection, sex and age information seem to have a relevant role in the diagnosis of Alzheimer’s disease. Considering this, this information has been added to the MRI volume used as input (in the data generator) when training the models in some experiments for comparison purposes. 

### 3.3. Proposed Solution

The aim of this work is to propose and develop a method to estimate the presence of Alzheimer’s disease by means of the analysis of brain MRI sequence. As it was explained previously in the Introduction Section, the severity of Alzheimer’s disease can be categorized through the CDR indicator used as a reference in the OASIS collection, whose value can be 0 (healthy), 0.5 (very mild), 1 (mild) or 2 (severe). Depending on the accuracy level, a two-, three- or four-class classification model can be developed. In this work, we have compared results of a two-class model and a three-class model, due to the CRD 1 and 2 categories containing fewer samples, so they have been combined.

The input of the model depends on the chosen architecture for the different tests. 2D (slice-level) and 3D (subject-level) approaches have been tackled. The 2D solution is usually used. The strongest point is that it allows to have a wider set of examples since there are 176 slices per MRI volume. This might allow a better training convergence and reduce overfitting. However, 2D approaches lack information in the third axis. This information when dealing with 3D objects, as it is the case with human tissues and organs, may provide details that will lead to a more precise solution [44].

For both 2D and 3D approaches, known architectures and custom architectures are considered. Among the known architectures, the ResNet [45] family has been evaluated, together with Inception [46], Xception [47], etc., with ResNet18 providing better results. Custom networks have been designed both in 2D convolution (Conv) and 3D convolution approaches. The implemented and compared network architectures are described next.

BrainNet2D (2D Conv): A small custom network is proposed. The input data is considered as a 3D subject-level with single output for the whole volume. Input data are M, N and K, with M and N being the size of every slice, and K the number of slices. However, the information is treated slice-by-slice through the 2D convolutional layers and max pooling layers in order to reduce spatial resolution and extract a representative number of high-level descriptors. Four convolutional blocks are included, each of them containing the 2D convolutional layer (filter size is 3) and the 2D max pooling layer. No additional fully connected layers are added. The activation function is ‘softmax’ for exclusive classes. The loss function is initially ‘categorical_crossentropy’. The architecture of BrainNet2D can be seen in Figure 1. Two variations of the same architecture are compared. First, the baseline BrainNet2D architecture (Figure 1A), and second, including Batch Normalization layers (Figure 1B) as an improvement technique, explained below.

BrainNet3D: A custom 3D network is proposed similar to the previous one but with a real 3D approach. Input data are M, N, K and 1, with M and N being the size of every slice, and K the number of slices. Five convolutional blocks are proposed, each of them containing the 3D convolutional layer and 3D max pooling layers. This approach fits better with biology and human understanding and might provide more useful patterns of structures in the brain. Filters for each convolutional layer represent the number of filters ∗filter size, with the number of filters being 8, 16, 32, 64 and 128 in each of the five convolutional blocks, and the filter size is 3 in all of them. At the end of the embedding part, a Global Average Pooling 3D layer is added. No additional fully connected layers are added. The activation function is ‘softmax’ for exclusive classes. The loss function is initially ‘categorical_crossentropy’. The architecture of BrainNet3D can be seen in Figure 1. Again, two variations of the same architecture are compared. First, the baseline BrainNet3D architecture (Figure 2A), and second, including the Batch Normalization layers as an improvement technique (Figure 2B).2D slice-level network: This network architecture performs at the 2D slice-level. This means that the network has as input a single slice of M, N and 3 size, and every image has an associated classification output. The unique channel per slice has been replicated to provide the three-channel input expected by ResNet family networks and to make it possible to fine-tune operations with ImageNet weights. ResNet18, a small model from the ResNet family [45] pretrained on ImageNet, has been used as a reference model. After the embedding part, and as in the previous networks, the activation function is ‘softmax’ and the loss function is ‘categorical_crossentropy’. In the testing phase, it must be pointed out that the final output for a complete study is provided in terms of ‘majority vote’ over all the slices of the sequence.

It is widely known by the people used to DL-based developments that there are some considerations in relation to the dataset and the specific details of the problem. If best practices are considered, the results may be improved [48]. The more remarkable strategies applied have been:

Cyclical Learning Rate: Cyclical Learning Rate (CLR) [49] is a strategy that allows oscillating between two learning rate values, iteratively. Previous traditional well-known learning strategies usually consider gradually decreasing the learning rate over the epoch using different functions (linear, polynomial, step, etc.). However, this strategy can lead the model to descend to areas of low loss values. With CLR, the optimal learning rate parameter can be easily found, and the model can consequently be better (and sooner) tuned. A minimum and a maximum learning rate value must be defined, and then the rate will cyclically oscillate between the two bounds. To do so, the different working policies can be defined: “triangular”, which is a simple triangular cycle, “triangular2”, also triangular but additionally cutting the maximum learning rate in half every cycle, and “exp_range”, which is similar to the previous but with an exponential decay. Our experiments with CLR revealed that the triangular policy reported the best results with our datasets.Batch Normalization: Batch Normalization [50] is a technique for training deep neural networks that standardizes the inputs to a layer for each mini-batch entering the network. This has the effect of stabilizing the learning process and often contributes to a better training process and thus better performance of the obtained model. Batch Normalization can also help to reduce overfitting, which is one of the main issues whenever a dataset is not large enough, as is the case with OASIS-1 and OASIS-2. This overfitting problem is mainly remarkable in the 3D-subject level approach. For the application of this Batch Normalization, the keras BatchNormalization() function, together with a ‘relu’ activation, is applied after the convolutional layer and before the max pooling layer. Figure 1 and Figure 2 illustrate how these layers have been integrated in the proposed BrainNet2D and BrainNet3D architectures, respectively. Metadata: Sex and age have been revealed as relevant variables for the diagnosis, as concluded in Appendix B. Therefore, these two variables could facilitate the establishment of the classification output and have been incorporated in the network [48]. The inclusion in the network has been carried out after the embeddings and before the final activation layer. The metadata has been considered as follows: two classes for sex (male, female), and two classes for age (<60 and >60). ImageNet: Some experiments, particularly the ones adopting the ResNet18 architecture, have used pre-trained ImageNet weights. This practice often improves the results and accelerates the training process.

## 4. Results

The results obtained for the proposed methodology are shown in this section. First, results for classification models over the OASIS-1 dataset and the results obtained for the classification models over the OASIS-2 dataset are shown for the two-class problem (cognitive normal vs. AD). Then, results for the three-class problem (cognitive normal vs. very-mild dementia vs. mild and moderate dementia) using the OASIS-2 dataset are presented. Metrics used for the evaluation of classification models are accuracy (ACC) and balanced accuracy (BAC).

Different trainings were carried out over the OASIS-1 dataset. Obtained results are shown in Table 2. All the experiments had common features that have not been included in the table. These common characteristics are the use of the horizontal planes of the 3D volume (coronal view), all the available MRI volumes have been used and the balanced inclusion of data according to sex has been also considered. These results correspond to the categorization of the input volume into healthy and with disease, i.e., a two-class classification problem. The table also shows the different parameters and operations that were used in the trainings, such as the normalization pre-processing action, experiment strategies (transfer learning, CLR triangular learning rate, Batch Normalization or the inclusion of metadata), the number of slices used in the process and the network image size. 

Results obtained for the classification models over the OASIS-2 dataset (obtained as explained in Appendix C) for the two-class problem are shown in Table 3. The same common characteristics as for classification models in OASIS-1 are applied here in all the trainings, however, experiments achieving worse results have been excluded from the comparison. 

The BAC metric is the one that reflects the real accuracy weighted by class, so it is the metric value that, in the opinion of the authors, better reflects the performance of the model. BrainNet3D with min–max scaling normalization and Batch Normalization, or BrainNet2D with [0, 1] normalization, CLR triangular learning rate and the inclusion of the sex and age as metadata, seem to show slightly better results. 

The experiments with the OASIS-2 dataset reveal improvements of the results for 2D network models. BrainNet2D- and ResNet18-based experiments provide BAC above 0.90, which is clearly above the state-of-the-art for these datasets. The BrainNet2D approach) achieves a BAC up to 0.92 as the average BAC of a 5-fold experiment, and the RestNet18 approach () a BAC up to 0.93. 

BrainNet3D with Batch Normalization also increases its performance without the help of the metadata. However, the improvement rate is not as high as for 2D approaches. The possible reason for the poorer improvement may be that for the subject-level input approach, that is, the 3D approach, the good skull removal and pre-processing applied to all the slices in the sequences is required in the input. Minor movements in the registration and inaccurate pre-processing operation over the OASIS-2 dataset performed by us, such as not perfect skull removal, may have led to incoherent and confusing input volumes for the 3D network. On the contrary, the 2D information of the slices is independent of the 3D volume—it does not matter whether there is good continuity on the z-axis. Every slice is dealt with separately, and the number of images is remarkably higher.

These additional experiments with the OASIS-2 dataset seem to reveal that the sex/age metadata are not a stable source of information to enrich the model during the classification task. The two experiments that included the metadata () achieved BAC values significantly inferior to the other experiments with the same model. This can be due to the fact that the metadata are highly biased by the characteristics of the population included in the datasets, but are not representative of the real clinical incidence of AD.

So far, a two-class problem has been addressed. Nevertheless, it is often useful to know the stage of the disease in case it is present. On that basis and relying on the best-performing models obtained from the experiments over the OASIS-2 dataset, a three-class problem has been tested. The experiments have been launched only over the OASIS-2 dataset due to the under-represented mild and severe stages of the disease in the OASIS-1 dataset. The results obtained are gathered in Table 4. 

These results again reveal that the 2D approach is more efficient than the 3D approach with subject-level input. BrainNet2D and ResNet18 solutions provide similar results, with the ones provided by ResNet18 being slightly better. These results point out that the methodology used is good to distinguish between mild Alzheimer’s and severe stages of the disease. This is really valuable, since the sooner the disease is detected the better, in order to start more adequate treatment. 

## 5. Discussion and Conclusions

The present work provided main contributions as described below. First, we performed a systematic literature review on the topic, where limitations of the different works also using the OASIS collection were addressed. The main drawback is always data scarcity to facilitate comparisons. In general, DL-based techniques and Convolutional Neural Networks usually require a big dataset to be properly trained, and especially for the complex problems, as this is. OASIS-1 is a commonly used dataset in works dealing with Alzheimer’s disease prediction. This dataset has strong points since it has well-registered and pre-processed sequences. Moreover, the skull is removed from the images and that ensures that the starting conditions for whatever study are the best. Masking for grey and white matters and cerebrospinal fluid is available, which makes it also useful to address segmentation problems. This is the reason why many works use this dataset in detriment of others that can provide a higher number of cases but that offer a poorly processed input, usually raw sequence data. Besides, the organization of the OASIS collection in different datasets, where images and annotations are easily accessible, facilitates the fair comparison of methods. 

OASIS-2 has been used for validation purposes of the proposed approach in a wider set of studies than the perfectly registered and pre-processed OASIS-1. However, the sequences in this dataset are provided in raw format. This includes artefacts, noise and other elements in the head, such as the skull, that require processing before using. It is advisable to make the input data of OASIS-2 as similar as possible to the OASIS-1 dataset. The main issue to be tackled is skull removal. In our work, a pre-processing pipeline has been proposed in Appendix C that aims at applying a similar pre-processing to the one semi-automatically performed in OASIS-1. In this way, additional OASIS-2 studies are available in the same conditions as OASIS-1 to be used in the same experiments. Unfortunately, both datasets could not be combined to improve model training since some of the subjects are repeated in both datasets and cannot be identified.

In this work, we proposed different approaches to tackle the problem: 2D and 3D networks in a slice-level approach or a subject-level approach. Custom networks (BrainNet2D, BrainNet3D) were proposed together with well-known architectures with transfer learning approaches such as fine-tuning with ImageNet weights. Modifications to the pre-processing have been applied to enhance the contrast of the grey-level structures in the brain. Different numbers of slices have been considered under the premise that it is not always the case that the higher the number of the images is, the better the results are. It might be true that it is worth providing a reduced number of valuable and meaningful slices instead of hundreds of slices that do not contain useful information. 

Table 2 and Table 3 showed results obtained for classification models over OASIS-1 and OASIS-2 datasets for a two-class classification problem (cognitive normal or with Alzheimer’s disease). Table 4 showed the results of a three-class (cognitive normal, very mild dementia, mild and moderate dementia) classification model that aims at obtaining higher precision in the detection of the stage of the disease. It can be shown that the presence of the disease can be established with a BAC of up to 0.93 with the OASIS-2 testing subset. In the three-class problem, additional stages in the disease can be predicted with a BAC of 0.88.

A comparison of these results with state-of-the-art-methods considered reproducible and comparable is shown in Table 5. Reference [28] proposed a ResNet model-based approach for feature extraction, where age and sex metadata are added as additional features and a support vector machine classifier is used for a three-class classification, achieving a BAC of 0.86. Reference [29] proposed a three-model ensemble using 112 × 112 crops input data from each anatomical plane and achieving an ACC of 0.93 for a three-class problem. Additionally, the authors of [31] performed a fair review of methods and strategies and their own implementation of them, reporting a BAC of 0.68 for a two-class problem using the OASIS dataset. 

From the observation of the table, we can conclude that the results obtained with the proposed solution are equal to the state-of-the-art or beyond in the case of two-class classification. To the best of our knowledge, the good results not only derive from the network architectures, but they also depend on the adequate strategies that have been implemented. The inclusion of the OASIS-2 dataset for validation of the proposed methodology has forced us to work in the MRI raw volume processing. This processing methodology can be validated as good since it has managed to generate good enough MRI volumes to provide good performance of the models. There are few works addressing the multiclass problem, with [28,29] being the most relevant ones. The results obtained by our method are very similar to the ones other authors have presented recently. We would like to point out that the metrics derived from our experiments are average metrics from 5-fold experiments, and not all the works used these metrics, but they present the best results obtained here. There are some experiments in this 5-fold approach that outperformed the state-of-the-art results, but we prefer to show the results as they are with average metrics of different experiments with randomly chosen training, validation and testing sets. This provides an idea about the stability and repeatability of the proposed solution. 

Future work clearly implies the use of additional datasets, such as ADNI, for comparative validation analysis and testing the generalization of the proposed approach. It is also desired to create a bigger combined dataset (from different sources) to increase the variability of the input samples of the different target classes, aiming to allow a better model generalization and reliable application to new unseen data. In this sense, a higher number of studies with high CDR will be useful, since they are currently under-represented. It would also be nice to dive deeper into three- and four-class problems to have the capability of predicting the different stages of the disease with high accuracy. This would lead to highlighting reliable diagnosis support tools that can aid in the definition of more accurate treatments and a hence impact the life expectancy of the patients. Other additional future research lines considered include studying the limits and weaknesses of the models in terms of accuracy and robustness [51], and adapting strategies such as multi-targeted backdoor [52], where models are led to misclassification using triggers on the data.

## Figures and Tables

**Figure 1 jpm-11-00902-f001:**
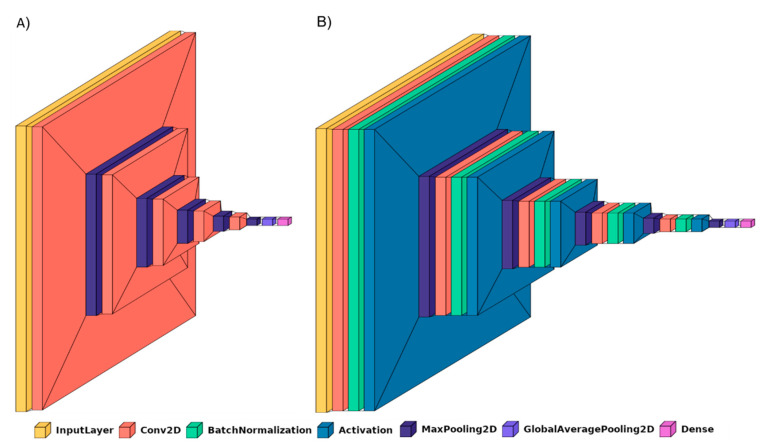
BrainNet2D network (**A**), and BrainNet2D network with Batch Normalization (**B**).

**Figure 2 jpm-11-00902-f002:**
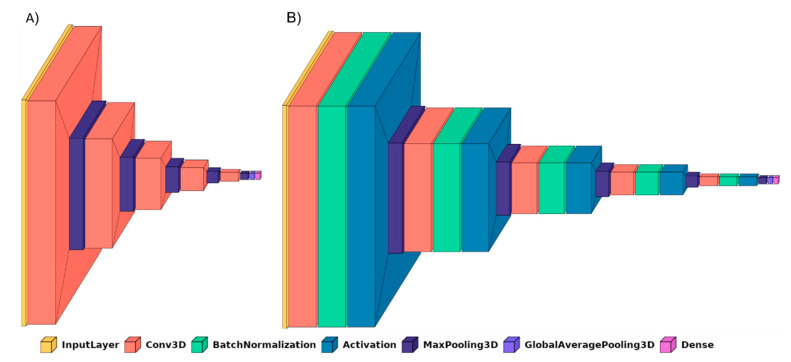
BrainNet3D network (**A**), and BrainNet3D network with Batch Normalization (**B**).

**Table 1 jpm-11-00902-t001:** Number of samples for the OASIS-1 and OASIS-2 datasets considering the CDR value as a reference.

CDR	OASIS-1	OASIS-2	OASIS-2 (Our Subset)
0 (cognitive normal)	336	206	177
0.5 (very-mild dementia)	70	123	98
1 (mild dementia)	28	41	27
2 (moderate dementia)	2	3	3
**TOTAL**	**436**	**373**	**305**

**Table 2 jpm-11-00902-t002:** Results for two-class classification, CDR = 0 (healthy) and CDR = 0.5, 1, 2 (disease), over the OASIS-1 dataset.

Network	Norm.	Strategies	Input Data	Slices Used	Image Size	Test ACC	Test BAC
**BrainNet2D**	[0, 1]	CLR triangular	3D	10 (centered in slice #88)	224	**0.81** [0.80, 0.80, 0.80, 0.85, 0.82]	**0.82** [0.81, 0.77, 0.78, 0.86, 0.86]
	min–max scaling	CLR triangular	3D	10 (centered in slice #88)	224	**0.81** [0.83, 0.80, 0.80, 0.82, 0.80]	**0.81** [0.80, 0.77, 0.77, 0.88, 0.82]
	[0, 1]	CLR triangular Batch Normalization	3D	10 (centered in slice #88)	224	**0.79** [0.77, 0.78, 0.86, 0.80, 0.72]	**0.79** [0.77, 0.78, 0.86, 0.80, 0.72]
	[0, 1]	CLR triangular Sex/Age metadata	3D	10 (centered in slice #88)	224	**0.79** [0.80, 0.77, 0.77, 0.88, 0.82]	**0.84** [0.83, 0.85, 0.85, 0.81, 0.85]
**BrainNet3D**	min–max scaling	None	3D	176 (all)	176	**0.78** [0.77, 0.78, 0.80, 0.78, 0.77]	**0.81** [0.83, 0.77, 0.77, 0.82, 0.85]
	min–max scaling	Batch Normalization	3D	176 (all)	176	**0.79** [0.82, 0.80, 0.78, 0.80, 0.77]	**0.83** [0.85, 0.79, 0.82, 0.86, 0.83]
	min–max scaling	Batch Normalization Sex/Age metadata	3D	176 (all)	176	**0.79** [0.78, 0.78, 0.83, 0.80, 0.78]	**0.82** [0.84, 0.77, 0.82, 0.86, 0.81]
	min–max scaling	Batch Normalization Sex/Age metadata CLR triangular	3D	176 (all)	176	**0.80** [0.80, 0.82, 0.85, 0.82, 0.74]	**0.84** [0.88, 0.80, 0.88, 0.88, 0.76]
**ResNet18**	[0, 1]	ImageNet weights	2D	10 (centered in slice #88)	224	**0.78** [0.80, 0.82, 0.77, 0.75, 0.75]	**0.79** [0.85, 0.79, 0.67, 0.80, 0.84]
	min–max scaling	ImageNet weights	2D	10 (centered in slice #88)	224	**0.81** [0.80, 0.80, 0.83, 0.80, 0.82]	**0.82** [0.80, 0.80, 0.83, 0.80, 0.82]
	min–max scaling	ImageNet weights CLR triangular	2D	10 (centered in slice #88)	224	**0.81** [0.82, 0.83, 0.86, 0.75, 0.77]	**0.83** [0.82, 0.81, 0.86, 0.80, 0.85]

**Table 3 jpm-11-00902-t003:** Results for two classes classification, CDR = 0 (healthy), CDR = 0.5, 1, 2 (disease) over OASIS-2 dataset.

Network	Norm.	Strategies	Input Data	Slices Used	Image Size	Test ACC	Test BAC
**BrainNet2D**	[0, 1]	CLR triangular	3D	10 (centered in slice #88)	224	**0.92** [0.94, 1.00, 0.92, 0.92, 0.83]	**0.92** [0.93, 1.00, 0.92, 0.92, 0.92]
	[0, 1]	CLR triangular Sex/Age metadata	3D	10 (centered in slice #88)	224	**0.82** [0.96, 0.93, 0.92, 0.46, 0.81]	**0.83** [0.96, 0.95, 0.92, 0.50, 0.80]
**BrainNet3D**	min–max scaling	None	3D	176 (all)	176	**0.67** [0.68, 0.80, 0.73, 0.51, 0.63]	**0.67** [0.68, 0.81, 0.73, 0.52, 0.64]
	min–max scaling	Batch Normalization	3D	176 (all)	176	**0.84** [0.81, 1.00, 0.78, 0.82, 0.79]	**0.84** [0.81, 1.00, 0.78, 0.83, 0.77]
	min–max scaling	Sex/Age metadata Batch Normalization	3D	176 (all)	176	**0.77** [0.70, 0.80, 0.82, 0.77, 0.77]	**0.78** [0.70, 0.81, 0.82, 0.79, 0.77]
	min–max scaling	Batch Normalization CLR triangular	3D	176 (all)	176	**0.79** [0.70, 1.00, 0.69, 0.87, 0.69]	**0.79** [0.70, 1.00, 0.68, 0.88, 0.70]
**ResNet18**	[0, 1]	ImageNet weights	2D	10 (centered in slice #88)	224	**0.92** [0.94, 1.00, 0.94, 0.92, 0.79]	**0.91** [0.94, 1.00, 0.94, 0.92, 0.77]
	[0, 1]	ImageNet weights CLR triangular	2D	10 (centered in slice #88)	224	**0.91** [0.94, 0.91, 0.94, 0.95, 0.83]	**0.92** [0.94, 0.94, 0.94, 0.94, 0.84]
	min–max scaling	ImageNet weights CLR triangular	2D	10 (centered in slice #88)	224	**0.93** [0.94, 1.00, 0.96, 0.92, 0.83]	**0.93** [0.94, 1.00, 0.96, 0.92, 0.81]

**Table 4 jpm-11-00902-t004:** Results for three-class classification, CDR = 0 (healthy), CDR = 0.5 (very mild stage) and CDR = 1, 2 (severe stage), over the OASIS-2 dataset.

Network	Norm.	Strategies	Input Data	Slices Used	Image Size	Test ACC	Test BAC
**BrainNet2D**	[0, 1]	CLR triangular	3D	10 (centered in slice #88)	224	**0.88** [0.94, 1.00, 0.78, 0.90, 0.77]	**0.85** [0.94, 1.00, 0.62, 0.92, 0.76]
**BrainNet3D**	min–max scaling	Batch Normalization	3D	176 (all)	176	**0.77** [0.87, 0.84, 0.80, 0.74, 0.58]	**0.76** [0.87, 0.85, 0.63, 0.82, 0.63]
**ResNet18**	[0, 1]	ImageNet weights CLR triangular	2D	10 (centered in slice #88)	224	**0.89** [0.94, 0.98, 0.84, 0.92, 0.79]	**0.88** [0.94, 0.99, 0.67, 0.94, 0.85]

**Table 5 jpm-11-00902-t005:** Comparison of the obtained results with the state-of-the-art methods that use the OASIS dataset for a two-class classification problem: Cognitive normal and Alzheimer’s disease. Comparison of results for a multiclass problem: Cognitive normal, Alzheimer’s disease in very mild state and Alzheimer’s disease in severe stage.

			CN vs. AD		Multiclass: CN vs. Mild vs. Severe	
Method	Approach	Dataset	ACC	BAC	ACC	BAC
(PuenteCastro, 2020) [28]	2D slice level	OASIS-1	--	--	--	0.86
(Islam and Zhang, 2018) [29]	2D slice level (112 × 112 crops)	OASIS-1		--	0.93	--
(Wen, 2020) [31]	2D slice level	OASIS-1 (over 62 years)	--	0.68 [0.68, 0.67, 0.69, 0.70, 0.66]	--	--
	3D subject level		--	0.68 [0.65, 0.70, 0.70, 0.71, 0.65]	--	--
**Our BrainNet2D**	2D slice level	OASIS-1	**0.79** [0.80, 0.77, 0.77, 0.88, 0.82]	**0.84** [0.83, 0.85, 0.85, 0.81, 0.85]		
**Out BrainNet3D**	3D subject level	OASIS-1	**0.80** [0.80, 0.82, 0.85, 0.82, 0.74]	**0.84** [0.88, 0.80, 0.88, 0.88, 0.76]		
**Our BrainNet2D**	2D slice level	OASIS-2	**0.82** [0.96, 0.93, 0.92, 0.46, 0.81]	**0.83** [0.96, 0.95 0.92, 0.50, 0.80]	**0.88** [0.94, 1.00, 0.78, 0.90, 0.77]	**0.85** [0.94, 1.00, 0.62, 0.92, 0.76]
**Out BrainNet3D**	3D subject level	OASIS-2	**0.84** [0.81, 1.00, 0.78, 0.82, 0.79]	**0.84** [0.81, 1.00, 0.78, 0.83, 0.77]	**0.77** [0.87, 0.84, 0.80, 0.74, 0.58]	**0.76** [0.87, 0.85, 0.63, 0.82, 0.63]
**ResNet18**	2D slice level	OASIS-2	**0.93** [0.94, 1.00, 0.96, 0.92, 0.83]	**0.93** [0.94, 1.00, 0.96, 0.92, 0.81]	**0.89** [0.94, 0.98, 0.84, 0.92, 0.79]	**0.88** [0.94, 0.99, 0.67, 0.94, 0.85]

## Data Availability

Data available in a publicly accessible repository that does not issue DOIs. OASIS (Open Access Series of Imaging Studies [13]).

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
