# Peer review of "MRI Deep Learning-Based Solution for Alzheimer’s Disease Prediction"

_jpm, 2021, doi:10.3390/jpm11090902_

Round 1
Reviewer 1 Report
This paper is worth for acceptance, novelty of the idea seems interesting and small changes need to be incorporated in order to enhance.
This paper deals with an exciting topic. The article has been read carefully, and some crucial issues have been highlighted in order to be considered by the author(s).
All the acronyms should be defined and explained first before using them such that they become evident for the readers.
The paper needs to be restructured in order to be precise.The Introduction and related work parts give valuable information for the readers as well as researchers. In addition recent papers should be added in the part of related work.
Representation of figures needs to be improved.
Grammatical errors should be validated.
It would be good if security domains [1], such as backdoor attack, would be reflected in future research or related work.
[1] Kwon, Hyun, Hyunsoo Yoon, and Ki-Woong Park. "Multi-targeted backdoor: Indentifying backdoor attack for multiple deep neural networks." IEICE Transactions on Information and Systems 103.4 (2020): 883-887.
Author Response
Please find answers in the enclosed document

Reviewer 2 Report
The authors extensive knowledge, as evidenced by their comprehensive literature review, has enabled their own groundbreaking research. The review includes: Alzheimer disease (AD) datasets and automated processing methods in the studies published during the period 2006-2016. Their selected strategy overcomes some of the pitfalls of other studies by using the proper analysis. They used the freely available OASIS-1 and OASIS-2 datasets. They used OASIS-1 to build a “gold standard” which then was applied to the OASIS-2 to validate the proposed approach. The authors are familiar with and used both fMRI analysis software –AFNI and FSL. Their impressive predictions show both the image-based automated diagnosis of AD and the establishment of the AD stage.
Minor:
line 802 – spelling of graey
Author Response

(The authors gave the same response as above.)

Reviewer 3 Report
In this manuscript, Saratxaga et al reviewed deep learning methods with MRI data for Alzheimer's disease prediction and proposed two potential methods. The authors straddled between a review, and a method article and made the manuscript much less readable. If the author could only focus on one aspect, for example, as a method paper, to spend more space with the clarification of the method and result would make the manuscript more clear.
Major points
- the title itself is not clear. There are many aspects of Alz’s disease prediction, and the authors only focused on the MRI-based prediction of Alz’s disease status.
- In the second part, the authors wanted to review some of the published methods. However, the authors only piled up a list of published methods and results. There are a plethora of very clear and informative reviews the authors could learn from including https://pubmed.ncbi.nlm.nih.gov/28087243/
https://pubmed.ncbi.nlm.nih.gov/28414186/
https://dl.acm.org/doi/fullHtml/10.1145/3344998or the authors can just remove the majority of this part, given the length of the manuscript. - Although the authors reviewed multiple data sets, the author only used a single data sets as input. I’m not sure about the generalizability of this method.
- The authors used extensive preprocessing of the image, which could make it difficult to apply to clinical usage. I’m not sure if the prepositioning will introduce a new bias. Given there are other datasets available (such as ADNI), can the model be validated by such data sets?
Minor Points
- Is the method also available to tell the difference between Alz and non-AD dementias?
- As a method paper, the authors should make the processing and modeling code available.
- Typo error discussion should be 5, not 4
Author Response

(The authors gave the same response as above.)

Round 2
Reviewer 3 Report
After revision, the manuscript looks much more readable than the previous version. It provides two potential solutions for the Alz disease status prediction with MRI data using deep learning methods. It would be great if the authors also release the code.